# Impact of Three-Year Intermittent Preventive Treatment Using Artemisinin-Based Combination Therapies on Malaria Morbidity in Malian Schoolchildren

**DOI:** 10.3390/tropicalmed5030148

**Published:** 2020-09-17

**Authors:** Hamma Maiga, Breanna Barger, Issaka Sagara, Abdoulaye Guindo, Oumar B. Traore, Mamadou Tekete, Antoine Dara, Zoumana I. Traore, Modibo Diarra, Samba Coumare, Aly Kodio, Ousmane B. Toure, Ogobara K. Doumbo, Abdoulaye A. Djimde

**Affiliations:** 1Institut National de Santé Publique, Bamako 1771, Mali; hmaiga@icermali.org; 2Malaria Research and Training Center, Department of Epidemiology of Parasitic Diseases, Faculty of Medicine and Dentistry, Faculty of Pharmacy, University of Sciences, Techniques and Technologies of Bamako, 1805 Bamako, Mali; isagara@icermali.org (I.S.); abdouguindo@icermali.org (A.G.); bila@icermali.org (O.B.T.); mtekete@icermali.org (M.T.); tonydara@icermali.org (A.D.); zouisaac@gmail.com (Z.I.T.); modibod@icermali.org (M.D.); scoumare@icermali.org (S.C.); alkodio@icermali.org (A.K.); obtoure@icermali.org (O.B.T.); 3Spokane Emergency Physicians, Spokane, WA 99201, USA; breannabarger@gmail.com

**Keywords:** malaria, morbidity, artemisinin-based combination therapy, IPTsc, Mali

## Abstract

Previous studies have shown that a single season of intermittent preventive treatment in schoolchildren (IPTsc) targeting the transmission season has reduced the rates of clinical malaria, all-cause clinic visits, asymptomatic parasitemia, and anemia. Efficacy over the course of multiple years of IPTsc has been scantly investigated. Methods: An open, randomized-controlled trial among schoolchildren aged 6–13 years was conducted from September 2007 to January 2010 in Kolle, Mali. Students were included in three arms: sulphadoxine-pyrimethamine+artesunate (SP+AS), amodiaquine+artesunate (AQ+AS), and control (C). All students received two full doses, given 2 months apart, and were compared with respect to the incidence of clinical malaria, all-cause clinic visits, asymptomatic parasitemia, and anemia. Results: A total of 296 students were randomized. All-cause clinic visits were in the SP+AS versus control (29 (20.1%) vs. 68 (47.2%); 20 (21.7%) vs. 41 (44.6%); and 14 (21.2%) vs. 30 (44.6%); *p* < 0.02) in 2007, 2008, and 2009, respectively. The prevalence of asymptomatic parasitemia was lower in the SP+AS compared to control (38 (7.5%) vs. 143 (28.7%); and 47 (12.7%) vs. 75 (21.2%); *p* < 0.002) in 2007 and 2008, respectively. Hemoglobin concentration was significantly higher in children receiving SP+AS (11.96, 12.06, and 12.62 g/dL) than in control children (11.60, 11.64, and 12.15 g/dL; *p* < 0.001) in 2007, 2008, and 2009, respectively. No impact on clinical malaria was observed. Conclusion: IPTsc with SP+AS reduced the rates of all-cause clinic visits and anemia during a three-year implementation.

## 1. Introduction

Intermittent preventive treatment (IPT) for malaria in children has recently been identified as an important tool in the fight against malaria, particularly in areas where malaria transmission follows a distinct seasonal pattern [1,2,3]. IPT, also referred to as seasonal chemoprophylaxis in areas with seasonal transmission, involves the administration of therapeutic doses of antimalarials to individuals in areas of high malaria transmission, regardless of their malaria infection status. Several clinical trials have demonstrated IPT efficacy in reducing morbidity and mortality associated with malaria infection [1,2,4,5,6,7,8,9]. The majority of these studies have focused on children, who are most vulnerable to the disease, and accordingly, a 2013 Cochrane review supported the implementation of IPT for infants and under-5 children living in areas with predominantly seasonal transmission [10].

In 2009, we reported the results of a randomized controlled trial of artemisinin-based combination therapies (ACTs) used for intermittent preventive treatment in school-aged children (IPTsc) [4]. The trial demonstrated a significant drop in the rate of clinical malaria, rate of asymptomatic parasitemia, and rate of anemia in those groups that received artemisinin combination therapy of sulphadoxine-pyrimethamine plus artesunate (SP+AS) or amodiaquine plus artesunate (AQ+AS). SP+AS conferred the greatest benefit. After the initial clinical trial, this cohort of schoolchildren was followed for two additional years to determine the reproducibility of these effects.

In areas of seasonal malaria transmission, IPTsc targeting the transmission season has decreased the rates of clinical malaria, asymptomatic malaria, all-cause clinic visits, anemia, and school absenteeism, but the efficacy of ACTs in the context of longitudinal IPTsc has been scantly investigated. Our study sought to understand the effects of longitudinal IPTsc among Malian schoolchildren. We hypothesized that IPTsc would continue to positively impact malaria morbidity reduction among school-aged children in an area with primarily seasonal transmission.

## 2. Materials and Methods

Trial design: This open, randomized controlled trial was performed among schoolchildren aged 6–13 years in Kolle, Mali. The study began in September 2007 and completed follow-up in January 2010. This three-armed study compared two artemisinin combination-based seasonal chemoprevention therapies (SP+AS or AQ+AS) against no seasonal chemoprevention therapy (control). Seasonal chemoprevention consisted of two doses given eight weeks apart, with the first dose administered at the beginning of back to school during the high seasonal malaria transmission. Children were followed monthly throughout the transmission season with physical exams, blood smears, and hemoglobin levels. Changes in the initial design included a protocol amendment for the no therapy arm. In the first year, the control (no therapy) arm received a single dose of vitamin C. In the second and the third years, the control arm only received water. This recapitulated a more accurate and pragmatic comparison group.

Participants: Kolle is a rural village of 3000 inhabitants located 57 km southwest of Bamako (the capital city of Mali). The study clinic was staffed 24 h a day, 7 days a week during the study follow-up period, serving as the only medical facility in the village. Based on our own epidemiological surveillance data, malaria is hyperendemic during the short transmission season from July to November. The prevalence of *Plasmodium falciparum* during the dry season is between 40% and 50%, which increases to 70–85% during the rainy season [11]. Inclusion criteria for enrolled participants consisted of matriculation in the village school, age of 6–13 years, absence of severe acute illness, ability to attend follow-up visits, written and expressed informed consent/assent from parent and student (10–13 years of age), respectively, and no history of allergy to study medications. Those not meeting all the inclusion criteria or with a history of chronic disease were excluded from the study.

Interventions: At the beginning of the study period, permission from village elders was obtained at a public meeting. Over a 5-day period, a village “crier” was used to recruit interested volunteers. The study was explained to children and their parents or guardian in the local language. Upon receiving the informed consent, students were assigned a computer-generated random number assigning them to one of three study arms—SP+AS arm: SP 25/1.25 mg/kg/day in a single dose plus artesunate 4 mg/kg once for 3 days; AQ+AS arm: amodiaquine 10 mg/kg/day plus artesunate 4 mg/kg/day for 3 days; control arm: vitamin C 250 mg tablet given once for 3 days in the year 2007. In the follow-up years 2 (2008) and 3 (2009), the control arm did not receive anything except water. Children remained in the same assigned study arm for all three years of the study. After randomization and arm assignments, children received an initial history and physical examination, where inclusion and exclusion criteria were reviewed. Any student with signs or symptoms of illness were treated according to the national guidelines. Finger-prick blood was collected for the preparation of thick and thin blood smears and the measurement of hemoglobin concentration (HemoCue^©^, Hemocue Inc., Brea, CA, USA). Blood smears were read by two independent certified readers. Conflicting results entered arbitration and were read by a third certified reader. This was considered the final read. Clinical malaria cases were defined as fever and constitutional symptoms (body aches, abdominal pain, headache, and vomiting) and the presence of parasites on the smear. Asymptomatic parasitemia was defined as study subjects not having signs of illness but having at least one parasite per 100 fields on microscopy. Anemia was defined according to the World Health Organization (WHO) [12]. All-cause clinic visits were defined as fever and constitutional symptoms (body aches, abdominal pain, headache, and vomiting) and the presence or absence of parasites on the smear. Children received their first course of study medication according to the arm assignment in September for 2007 and 2008 and in October for 2009. At the time of the initial treatment dose, researchers were aware of the arm assigned for each student, and all doses were directly observed. Participants were observed for signs of intolerance for at least 30 minutes after each dose. If vomiting occurred within 30 minutes, the treatment was re-administered. Children returned on days 2 and 3 to receive the remaining therapy doses if assigned to either the SP + AS or AQ + AS arms.

Outcomes: The primary outcome was the number of episodes of clinical malaria. Secondary outcomes were asymptomatic parasitemia, hemoglobin concentration, all-cause clinic visits, and school performance, as observed by their classroom teacher. In order to measure these endpoints, the team performed monthly follow-up visits for the 5 months following the initial treatment dose. Follow-up visits included history, physical examination, and finger-prick blood collection (parasitemia and hemoglobin concentration measurement). Students received a second dose of study medication 8 weeks after the initial course. In addition to the active monthly follow-up, participants were instructed that they should come to the clinic for any illness. All unscheduled visits by participants to the clinic during the study period were documented. Children with signs and symptoms of malaria underwent thick and thin smears and hemoglobin testing. In years 1, 2, and 3, students with clinical malaria were treated with the same IPTsc treatment. Students in the control arm received SP monotherapy in year 1 and were treated with artemether-lumefantrine (Coartem^®^) in years 2 and 3. Cases of severe malaria, defined by WHO criteria in 2000, were treated with quinine. Clinicians provided supportive care and treatment for all other diagnoses according to national guidelines.

Sample size: Based on our previous studies in Kolle, we expected the incidence of malaria infection in the control arm to be 75%. Seeking to detect a 33% reduction in malaria infection with a power of 95%, an alpha risk of 5%, and allowing 10% loss to follow-up, 97 children were required in each arm for a total sample size of 291 children. Because children leave Kolle after the sixth grade to attend middle school in a neighboring village, children that graduated from the sixth grade were exited from the study. Therefore, approximately 50 students were intentionally exited from the study each year.

Randomization sequence generation and statistical methods: Participants were randomized using a computer-generated shuffle (Excel^©^, Microsoft Inc., Washington, WA, USA) of participant study numbers and corresponding medication group. As children presented for enrollment, they were assigned a study number in the numerical order of presentation within a given grade. Participants were stratified by grade level such that there were approximately equal numbers of each study arm represented. Grade level was defined by the success of the new classroom. We had 6 classrooms (1st, 2nd, 3rd, 4th, 5th, and 6th). School grade average was the annual mean obtained by the sum of the all-month grade divided by the total number of months. Clinicians were aware of drug assignment at the time of initial randomization. Data were double entered using Microsoft ACCESS (Microsoft Inc., Washington, WA, USA), and the statistical analysis was performed using R software. Baseline characteristics of the volunteers among treatment arms versus control were compared using the Fisher Exact test and Chi-square tests for categorical variables, Student for continuous variables, and the Mann–Whitney *U*-test/Wilcox test for non-normally distributed continuous variables. The McNemar’s test and the Paired sample *t*-test were used before and after each treatment arm. For binomial outcome, a weighted logistic regression was used for multivariate analysis (anemia, malaria), and for continuous outcomes (hemoglobin concentration, parasitemia, grade average). Statistical significance was set at *p* < 0.05.

Ethics: The study protocol was reviewed and approved by the Ethical Committee of the Faculty of Medicine and Dentistry, Faculty of Pharmacy/University of Sciences, Techniques and Technologies of Bamako (USTTB), Mali. Project Identification Code: 59_FMPOS.

### 2.1. Participant Flow

Out of a total of 475 students aged 6 to 13 years in the village, 305 were screened, and 296 were enrolled. The total loss to follow-up was two (0.7%) students at the end of the initial trial period in January 2008 (Figure 1). One student was excluded from school for being less than 6 years old, and the other refused follow-up at the second treatment dose; both students were from the SP + AS arm. In 2008 and 2009, 53 and 59 students, respectively, were aged out of the trial. Because this was our intended study design, they did not count toward loss to follow-up.

### 2.2. Baseline Data

Baseline characteristics, including age, hemoglobin concentration, and parasitemia, were similar between the study arms at the start of the study in 2007 (Table 1). Cohort baseline follow-up numbers dropped to 243 in 2008 and 184 in 2009, resulting from children aging out of the study.

### 2.3. Clinical Illness

A total of 144, 92, and 66 unscheduled clinic visits occurred in 2007, 2008, and 2009, respectively. Of these visits, 99 (68.8%), 36 (39.1%), and 15 (22.7%) had malaria confirmed by thick smear, respectively, in 2007, 2008, and 2009. There were no cases of severe malaria. The control arm experienced the most episodes of all-cause clinic visits, with 47.2%, 44.6%, and 44.6% during the season compared to 20.1%, *p* < 0.001; 21.7%, *p* = 0.005; and 21.2%, *p* = 0.02 in the SP+AS arm and 32.6%, *p* = 0.009; 33.7%, *p* = 0.14; and 33.7%, *p* = 0.56 in the AQ+AS arm in 2007, 2008, and 2009, respectively (Table 2). Children having smear-confirmed cases were 36.1%, 12.0% and 6.1% in Control arm compared to 12.5%, 10.9% and 6.1% in SP+AS arm and 20.1%, 16.3% and 10.6% in AQ+AS arm in 2007, 2008, and 2009, respectively. This difference was significant only in year 1 of the trial, after which a downward trend across all groups (in terms of % children experiencing clinical malaria) was observed.

### 2.4. Parasitemia

Parasite detection in years 2 and 3 varied from our initial observations in year 1 of the trial (Figure 2). After the first dose in year 1 (2007), parasite prevalence was significantly decreased in the SP+AQ and AQ+AS arms compared to the control arm (*p* < 0.001). In November, we observed an initial downward trend in parasite load for groups receiving IPT, with a subsequent convergent spike at 8 weeks post-initial IPT dose (*p* > 0.05). In December and January, this difference was significantly decreased after the second dose in SP+AS and AQ+AS arms compared to the control arm (*p* < 0.001). After the first dose in year 2 (2008), parasite prevalence was significantly decreased in the SP+AS arm only (*p* = 0.003). In November, the peak occurred at 4 weeks post-intervention (rather than 8 weeks post-intervention) in all three arms (*p* > 0.05). In December, this difference was significantly decreased in SP+AS and AQ+AS arms compared to the control arm (*p* = 0.003), and no difference was observed in January. In year 3 (2009), the beginning of the seasonal chemoprophylaxis dosing was postponed due to a delayed school start. Thus, volunteers began the season with higher initial parasite loads. IPT lowered parasite loads for the SP+AS arm (*p* = 0.007), but not the AQ+AS arm (*p* = 0.47) compared to the control arm; a slight rise in parasitemia rates was also noted. Besides, the significant decrease of parasite prevalence in the control arm compared to the AQ+AS arm (*p* = 0.04) before treatment was also noted. By the end of the season (December, January, and February), however, all arms had lower parasite loads than at any point during the season. Overall, volunteers receiving SP+AS trended towards the lowest rate of parasite load compared to the AQ+AS and control arms at the end of the season, though this was not a statistically significant finding (*p* > 0.05) (Figure 2). *Plasmodium* species identified in this study were *Plasmodium falciparum*, *Plasmodium malariae*, and *Plasmodium ovale* (results not shown).

### 2.5. Plasmodic Index

Plasmodic index (PI) by month (M) for all the years was 14%, 29%, 30%, 15%, and 12% in month 1 (M1), M2, M3, M4, and M5, respectively (Figure not shown). There was a slight increase in parasite prevalence in M2 and M3 (*p* < 0.001), and it significantly decreased in M4 and M5 (*p* < 0.001) compared to M1. Globally, PI was 48%, 36%, and 16% in 2007, 2008, and 2009, respectively. The difference was significantly decreased from 2007 to 2009 (*p* < 0.001) but not from 2008 to 2009 (*p* > 0.05) (Figure not shown).

In Appendix A, the PI of SP+AS arm was significantly decreased in M2 compared to M1 (*p* = 0.012), significantly increased in M3 (*p* = 0.002), and significantly decreased in M4 and M5 (*p* < 0.001). For the AQ+AS arm, PI was significantly increased in M2 and M3 (*p* = 0.007 and *p* = 0.01) and decreased in M4 and M5 (*p* = 0.01) compared to M1 (Appendix A). For the control arm, PI was significantly increased in all months compared to M1 (*p* < 0.001) (Appendix A). In M1, the PI was similar in the three arms (*p* > 0.05). In M2, the PI was significantly different in the SP+AS and AQ+AS vs. control (*p* < 0.004), but this difference disappeared in M3 (*p* > 0.05) (Appendix A). This difference was significantly decreased in SP + AS and AQ + AS arms compared to the control arm in M4 and M5 (*p* < 0.001).

In year 2007, the PI was significantly decreased in the SP+AS arm 38 (7.5%) and AQ+AS arm 60 (11.7%) versus Control arm 143 (28.7%), *p* < 0.001 (Appendix A). In 2008, The PI was significantly decreased in the SP+AS arm 47 (12.7%), *p* = 0.002 but not AQ+AS arm 62 (16.2%), *p* = 0.08 vs. control arm 75 (21.2%). No difference was observed between any of the three arms in 2009, *p* > 0.05 (Appendix A)

Parasitemia level in all arms was similar in M1 (*p* > 0.05, Appendix A). In M2, parasitemia was significantly decreased in the SP + AS arm compared to the control arm (*p* < 0.001) but increased in the AQ + AS arm compared to the control arm (*p* = 0.007) (Appendix A). In M3, parasitemia was similar in all arms. In M4 and M5, parasitemia was significantly decreased in the SP + AS and AQ + AS arms compared to the control arm (*p* < 0.001) (Appendix A). Globally, a difference of parasitemia was noted from the year 2007 to 2008 (*p* > 0.05), but this difference was significantly decreased from 2007 to 2009 (*p* < 0.001) (Appendix A).

### 2.6. Anemia

Globally, at follow-up for all doses, children treated with IPT were significantly less likely to be anemic (SP + AS 29.9%, 29.1% and 13.4%, *p* = 0.001 versus control 42.1%, 44.6% and 25.3% in 2007, 2008, and 2009, respectively, and AQ + AS 30.6%, *p* < 0.001 and 35.5%, *p* = 0.02 vs. control 42.1% and 44.6% in 2007 and 2008, respectively, Table 3). There were no cases of severe anemia. At the end of the second dose, IPT arms were significantly less likely anemic (SP+AS arm vs. control arm in 2007 and 2009, *p* = 0.04 and AQ+AS arm vs. control arm in 2007, *p* < 0.002; Appendix A). This difference was significant after the first dose in 2008 with SP+AS arm vs. control arm (*p* = 0.001). At the end of the study, the control arm continued to have higher rates of anemia; this difference was significant in the SP+AS arm in 2007 (*p* = 0.01) and not significant in other years. From the first month to the last month, the incidence of anemia was significantly decreased in the SP+AS arm in 2007 and 2009 (*p* < 0.006), in the AQ+AS arm in 2009 (*p* < 0.003), and in the control arm in 2009 (*p* < 0.01), excluding January (Appendix A). The global prevalence of anemia was 45.5%, 33.8%, 33.1%, 24.0%, and 34.5% in Month 1 (M1), M2, M3, M4, and M5, respectively (Figure not shown).

### 2.7. Hemoglobin Concentration

Hemoglobin concentration levels at each evaluation are shown in Appendix A. In the SP+AS arm, levels were 11.96 g/dL (95% CI: 11.89−12.03), 12.06 (95% CI: 11.98−12.15), and 12.63 g/dL (95% CI: 12.27−12.52); in the AQ+AS arm, levels were 11.91 g/dL (95% CI: 11.98−12.05), 11.92 g/dL (95% CI: 11.84−11.99), and 12.26 g/dL (95% CI: 12.15−12.36); in the control arm, levels were 11.60 g/dL (95% CI: 11.53−11.67), 11.64 g/dL (95% CI: 11.56−1172), and 12.16 g/dL (95% CI: 12.05−12.27) in 2007, 2008, and 2009, respectively. The difference was statistically significant in SP+AS (11.96, 12.06, and 12.62 g/dL; *p* = 0.001) (Appendix A), and AQ+AS (11.91, *p* < 0.001; 11.85, *p* < 0.006; and 12.25 g/dL, *p* = 0.46) (Appendix A) vs. control 11.60, 11.64, and 12.15 g/dL in 2007, 2008, and 2009, respectively.

### 2.8. School Performance

The school grade average in the SP+AS arm (5.48, 4.84, and 5.00) and in the AQ+AS arm (5.49, 4.66, and 4.93) was compared to the control arm (5.00, 4.40, and 4.65); *p* > 0.05, in 2007, 2008, and 2009, respectively. The success of different treatment arms was higher in all years (> 77.9%) (Table 4). The successes of schoolchildren were 91% (21), 95% (20), and 79% (15) in the SP+AS arm; 91% (20), 86% (19), and 78% (14) in the AQ+AS arm; 85% (13), 85% (11), and 71% (10) in the control arm in 2007, 2008, and 2009 (*p* > 0.05), respectively (Table 4). The grade level of SP+AS and AQ+AS arms was higher than the control arm in all years but not significant *p* > 0.05 (Appendix A).

### 2.9. Adverse Events

Adverse events were defined as fever, headache, abdominal pain, respiratory symptoms, vomiting, diarrhea, lethargy, and pruritus. The most frequent were fever, headache, abdominal pain, and respiratory symptoms. Fever and headache were significantly decreased in the SP+AS arm (1.2%, 2.6%) vs. control arm (2.7% and 3.8%); *p* = 0.001 and *p* = 0.04, respectively. Breathing problems were significantly increased in the SP+AS arm (1.4%) vs. control arm (0.5%), *p* < 0.001 (Table 5). Across all groups, headache was the most commonly reported side effect for both rounds of treatment. The rate of adverse events was 52/919 (5.7%), 28/633 (4.4%), and 21/483 (4.3%) for the SP+AS arm; 43/954 (4.5%), 44/700 (6.3%), and 25/425 (5.9%) for the AQ+AS arm; 52/939 (5.5%), 33/611 (5.4%), and 24/461 (5.2%) for the control arm in 2007, 2008, and 2009, respectively; this difference was not significant *p* > 0.05 (Table not shown). No students elected to quit the study because of these events.

## 3. Discussion

School-aged children suffer consequences of malaria but are generally neglected in malaria control strategies. One control strategy under investigation is IPT in school children (IPTsc) [4]. IPT with SP is recommended in pregnancy (IPTp) and in infants (IPTi) living in areas with moderate to high malaria transmission. Seasonal malaria chemoprevention (SMC) with monthly SP+AQ is recommended in children aged < 5 years living in areas with highly seasonal malaria transmission in the Sahel sub-region of Africa [3]. In recent years, significant efforts have been made to effectively prevent malaria among African children [13].

Our results suggest that IPTsc administration of SP+AS would benefit schoolchildren living in areas with high malaria endemicity.

In this controlled, randomized trial of school-aged Malian children living in a high-transmission setting, IPTsc with either SP+AS or AQ+AS was superior to control in reducing malaria incidence (year 1), all-cause clinic visits, anemia, and asymptomatic parasitemia [4], though data and strength of the association varied over successive years. IPTsc with SP+AS appears to be safe and well-tolerated based on our study data. We collected height and weight data on all participants and found no evidence that IPTsc negatively impacted growth. These findings, along with other previous studies [4,14,15,16], have shown the potential of IPT to improve school attendance by preventing acute illness and reducing anemia. Malaria incidence was significantly reduced in both SP + AS and AQ + AS arms in 2008 and 2009, though this effect was probably associated with increased student age and the small size of the study.

Protection against anemia and all-cause clinical visits was observed in the SP + AS arm, but asymptomatic parasitemia appeared to be short-lived, though comparisons varied by year and regiment. Although each treatment arm was superior to the control one month after the first dose, this difference was lost after two months, suggesting that protection waned between 4 to 8 weeks post-treatment. For IPT, a drug should eliminate circulating parasites and persist at sufficient levels to prevent the multiplication of parasites acquired between doses. This time frame is consistent with previous findings of an IPT trial in Ghanaian infants, in which protection waned between 5 and 6 weeks post-treatment [17]. In our study, IPT with SP+AS was highly efficacious in reducing parasitemia and preventing malaria compared to the control in 2007 and 2008. These findings were consistent with the long half-life of sulfadoxine and pyrimethamine [18]. When parasites are sensitive to the drug, this is an advantage because long-term prophylaxis is conferred. Therefore, SP + AS performed better than AQ+AS in our study.

Previous studies have shown that SP is highly efficacious for the treatment of malaria, with substantial post-treatment prophylactic efficacy in West Africa [19,20,21]. However, resistance to SP is rapidly increasing throughout much of Africa [11,22,23], which may reduce its effectiveness for use in IPT. Furthermore, there is some evidence that IPT with SP may encourage the development of resistance [24], although data are conflicting [13]. IPTsc with SP + AS significantly improved hemoglobin levels and reduced the risk of anemia, probably due to effects against malaria. In a previous Ugandan study, IPTsc with dihydroartemisinin-piperaquine (DHA-PQ) had no effect on anemia, and the parasite prevalence was 17.8% in children receiving IPT with a placebo, compared with 12.0% in our study [16]. The explanation for differential impacts on parasitemia and anemia in the two studies may be due to the differences of transmission intensities, the drugs used for IPT, or the socioeconomic status of study participants.

SP resistance associated with AQ in SMC is a reality [25], while selection of resistance to artemisinin has been reported [26,27,28,29]. It remains to be determined whether IPTsc using ACTs will affect the development of drug resistance and the future efficacy of these critically important drugs. The matched elimination half-life for AQ and SP would probably result in protection by the partner drugs in the combination.

This study detailed the effects of longitudinal IPTsc with SP+AS and AQ+AS arms. During year 1, the effects were quite dramatic. While there continued to be a positive downward effect on the number of cases of malaria and asymptomatic parasitemia, the effect size declined over the two following years. This is likely attributable to the fact that the follow-up group consisted of older children, on average, as no new children were added to the study after initial enrollment. Another explanation could be that there were fewer children carrying parasite loads as much of the population had received IPT the years prior. Moreover, the declining efficacy could also be due to an overall decrease in transmission intensity in the area.

Our most notable challenges were financial and political issues that arose during our trial. We were unable to obtain scholastic indicators before the final year of follow-up. Moreover, our dosing calendar was altered because of a teacher strike resulting in delayed dosing of several weeks. While this makes the data more difficult to compare year to year, it does give a more practical and real-world analysis of what a school-linked IPT program could expect to achieve under typical circumstances. Although protection from clinical malaria and reduced anemia is expected to result in improved school attendance and performance [15,30], these were not directly measured in this study. We observed a higher grade average in the SP + AS and AQ + AS arms compared to the control arm, but these differences were not statistically significant.

One additional change to protocol occurred after year 1 (2007), in which the control group received SP alone for treatment of symptomatic malaria, but during the years 2 and 3, they were treated by artemether-lumefantrine (Coartem^®^); it is possible that ACT impacted the control group in 2008 and 2009. Finally, IPTsc implementation in our study was also impacted by the initiation of indoor residual spray (IRS), insecticide-treated nets (ITNs), and long-lasting insecticide-treated nets (LLINs); these strategies provide protection against morbidity and mortality attributable to malaria [31,32,33].

## 4. Conclusions

IPTsc with SP+AS reduced the rates of all-cause clinic visits and anemia in three-year implementation together with IRS, ITNs, and LLIN strategies in Kolle.

## Figures and Tables

**Figure 1 tropicalmed-05-00148-f001:**
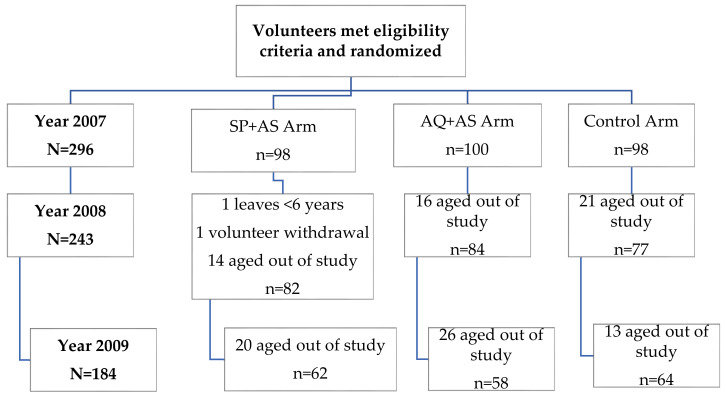
Study profile.

**Figure 2 tropicalmed-05-00148-f002:**
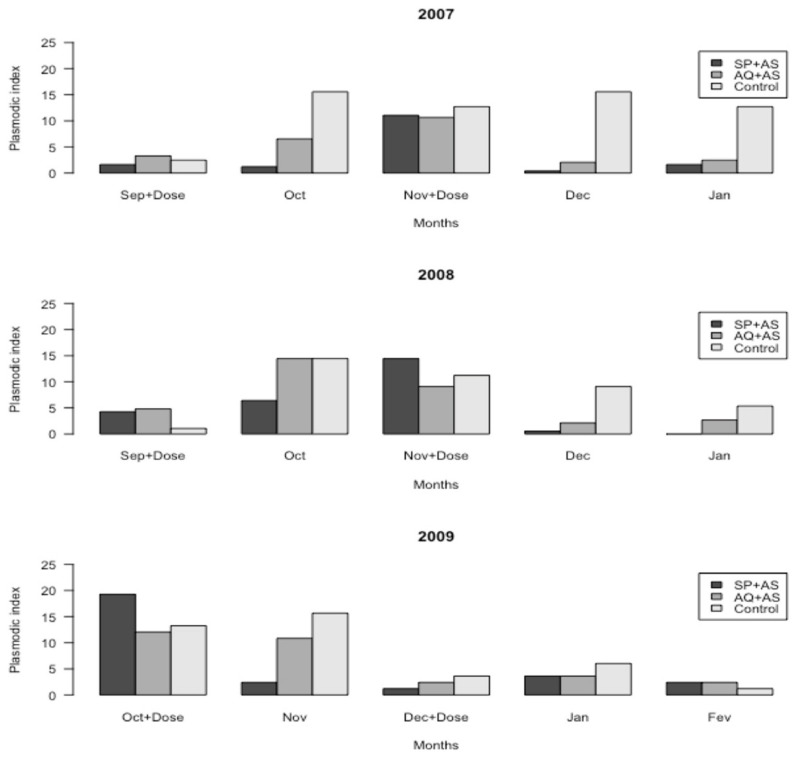
The prevalence of *Plasmodium* sp. in the treatment arms—monthly and by years.

**Table 1 tropicalmed-05-00148-t001:** Baseline Characteristics.

Parameters	Treatment Arms		
	SP+AS (N = 98)	AQ+AS (N = 100)	Control (N = 98)
Mean age, years (range)	10 (6 – 13)	10 (6 – 13)	10 (7 – 13)
Percent female %	36.3	46.6	42.9
Mean weight, kg (range)	27 (12–55)	26 (15–55)	28 (16–48)
Mean hemoglobin, g/dL (SD)	11.4 (+/−1.4)	11.3 (+/−1.5)	11.2 (+/−1.4)
Geometric mean parasites/µL (SD)	2107 (+/−5334)	2627(+/−8123)	2407(+/−2073)
Axillary temperature (°C)	36.5	36.6	36.5

kg = kilogram; g/dL = grams per deciliter; SD = standard deviation.

**Table 2 tropicalmed-05-00148-t002:** Unscheduled Visits and Malaria Episodes.

		Treatment Arms				
		SP+AS n (%)	*p*-Value	AQ+AS n (%)	*p*-Value	Control n (%)
2007	All-cause clinic visits	29 (20.1)	<0.001	47 (32.6)	0.04	68 (47.2)
	Malaria cases	18 (12.5)	<0.001	29 (20.1)	0.009	52 (36.1)
2008	All-cause clinic visits	20 (21.7)	0.005	31 (33.7)	0.14	41 (44.6)
	Malaria cases	10 (10.9)	0.98	15 (16.3)	0.63	11 (12.0)
2009	All-cause clinic visits	14 (21.2)	0.02	22 (33.7)	0.56	30 (44.6)
	Malaria cases	4 (6.1)	1	7 (10.6)	0.36	4 (6.1)

Comparison of the number of all-cause clinic visits and clinical malaria cases by year and study treatment arm. The test statistic (*p*-value) calculated by Chi-square statistic, comparing the intervention group to the control.

**Table 3 tropicalmed-05-00148-t003:** Comparison of Anemia Prevalence by Treatment Arms in Each Study Year.

		Treatment Arms				
		SP+AS	*p*-Value	AQ+AS	*p*-Value	Control
2007	n (%)	116 (29.9)	<0.001	122 (30.6)	<0.001	162 (42.1)
2008	n (%)	80 (29.1)	<0.001	100 (35.5)	0.02	122 (44.6)
2009	n (%)	29 (13.4)	0.001	36 (18.8)	0.12	55 (25.3)

Comparison of the number of anemia cases by year and study treatment arm. The test statistic (*p*-value) was calculated by Chi-square statistic, comparing the intervention group to the control.

**Table 4 tropicalmed-05-00148-t004:** Markers of School Performance in Each Study Year in The Treatment Arms.

Treatment Arms	Markers	2007−2008	2008−2009	2009−2010
SP + AS	Grade average	5.48	4.84	5.00
	95% CI	(4.93–6.03)	(45–5.24)	(4.71–5.27)
	n	23	21	19
	*p*-value Success (%)	0.49 91	0.41 95	0.38 79
AQ + AS	Grade average	5.49	4.66	4.93
	95% CI	(5.07–5.91)	(4.26–5.05)	(4.53–5.33)
	n	22	22	18
	*p*-value Success (%)	0.41 91	0.64 86	0.57 78
Control	Grade average	5.00	4.40	4.65
	95% CI	(4.50–5.49)	(4.00–4.81)	(4.30–5.00)
	n Success (%)	15 87	13 85	14 71

Comparison of grade average by year and study treatment arm. The test statistic (*p*-value) was calculated by *t*-test independent statistic, comparing the mean of the intervention arm to the control arm. Comparison of the number of successes by year and study treatment arms. The test statistic (*p*-value) was calculated by Chi-square statistic, comparing the intervention arms to the control.

**Table 5 tropicalmed-05-00148-t005:** The Proportion of Adverse events of All Years in School-Aged Children During Two Rounds of IPTsc Treatment Using Passive and Active Surveillance.

Treatment Arms								
Adverse Events	SP + AS			AQ + AS			Control	
	n	(%)	*p*-Value	n	(%)	*p*-Value	n	(%)
Fever	25	1.2	0.001	53	2.5	0.77	54	2.7
Headache	54	2.6	0.04	79	3.8	0.97	77	3.8
Vomiting	24	1.2	0.15	35	1.7	0.89	35	1.7
Abdominal pain	5	1.3	0.08	14	1.7	0.45	12	2.0
Coughing	28	1.4	<0.001	5	0.2	0.42	9	0.5
Diarrhea	5	0.3	0.12	14	0.7	0.72	12	0.6
Pruritis	2	0.1	1	2	0.1	1	1	0.05
Lethargic	0	0	-	1	0.04	-	0	0
Convulsion	0	0	-	0	0	-	0	0

Comparison of the number of adverse events by all years and study treatment arms. The test statistic (*p*-value) calculated by the Chi-square test and Fisher’s Exact test, comparing the intervention arms to the control arm.

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
