# Peer review of "Impact of Three-Year Intermittent Preventive Treatment Using Artemisinin-Based Combination Therapies on Malaria Morbidity in Malian Schoolchildren"

_tropicalmed, 2020, doi:10.3390/tropicalmed5030148_

Round 1
Reviewer 1 Report
The study seens to be well performed and I believe the results are of interest of the general public. However, several minor corrections are necessary:
line 20: was scantily investigated.
line 24: respect to the incidence
line 25 to 30: please rewrite, it is very confusing...
line 41: children, the most vulnerable
line 42: disease and, accordingly, a 2013
line 43: and children under-5 years
line 50: the reproducibility of these effects.
line 52: has decreased the rates
line 59: trial was performed among school
line 60: completed the follow-up
line 66: Changes in the initial
line 67: In the first year,
line 68: In the second and the third years,
line 76: age of 6–13 years
line 78: from the parents and the student OR from the parents or the student?
line 79: studied medications. . Those not meeting all the inclusion
line 81: At the begining of the
line 84: receiving the informed consent
line 87: 2007. In the
line 89: for all the three years
line 91: where the inclusion
line 97: (and OR or?) the presence of parasites on smear.
line 99: according to the World
line 101: At the time of the
line 102: assigned for each student
line 104: Children returned on days 2 and 3 for directly observed therapy doses (what do you mean?)
line 106: was the number of episodes
line 116: Children with signs and symptoms
line 158: 158 between the study arms
line 183: the peak occurred at 4 weeks
line 184: all the three arms
line 198: a slight increase in the
line 237: evaluation are shown
line 260: lethargy and pruritus. The most frequent were
line 262: Breathing problems were significantly increased
line 268: No students were elected
line 276: chemoprevention was achieved
line 279: at the peak of malarial transmission
Reviewer 2 Report
The manuscript presents a well-thought-out study on the importance of intermittent preventive treatment for malaria. Although the study covered most aspects to determine the effect of ITPs in local settings and manuscript is comprehensive, I am not sure of the relevance of this study in current time given a decade has passed since the study was conducted.
- The prevalence of malaria, and resistance against sulphadoxine-pyrimethamine and ACTs might be different from what it was 10 years ago. The authors need to provide an explanation in the discussion if the study is still relevant in today’s context and what is the current status of antimalarial resistance observed in Mali.
- There are many grammatical mistakes throughout the manuscript. The authors need to go through the manuscript and correct the errors
- The abstract does not mention the results for Amodiaquine+artesunate (AQ+AS). The authors should consider mentioning the outcome for this arm of the study as well.
- The authors should consider discussing why (AQ+AS) was not found to be equally effective as (SP+AS)
- The authors should also provide an insight into the current policy with regards to treatment and prophylaxis of malaria in school children in Mali? The study is out-of-date and must be tied in with the current times. It has been 10 years since the study was conducted and there are other similar studies in support of ITP, so has the local government adapted ITPs as a policy to control malaria infection in school children? If so, which ITP has been adapted? If not, why they have not done so? Is there any conflicting evidence which has emerged in the past 10 years? The authors should consider discussing this as well.
Reviewer 3 Report
This is an interesting article. I have a few suggestions, which can be considered to revise the manuscript.
- The authors could also compare SP+AS and AQ+AS arms for variables presented in table 2 as well as plasmodic index, anemia prevalence, hemoglobin and school performance to know which combination is better.
- Line 139: typing error should be corrected "he Paired sample t-test".
- Line 226: replace inn with "in"
- There are no figures 6a and 6b but they have been cited in lines 242-243.
- The authors should be uniform with the UK or US English (hemoglobin or haemoglobin; anaemia or anemia etc) as per the journal requirement in the manuscript as well as in the suppl.
- Some English editing in the manuscript is needed to reflect a clear message. 1) Success of all years was higher (Table 4) compared to ??. 2) Therefore, the grad level was more than of IPT arms compared to Control arm in all years but not significant p>0.05 (Figure S4) etc.
- replace 1,2 with 1.2 in table 5
- Figure S3 is available in the suppl but not cited in the manuscript.
- The number of malaria cases presented in table 2 can be influenced by the amount of rain, traveling and living sites and conditions of participating children? if yes, so how did authors control for these and other confounding factors?
- The authors should clarify which Plasmodium species were identified (Figure 2).
- Be uniform, figure or fig.
- Being clinical malaria as a primary outcome did authors identify if it was due to reinfections or recrudescences in the following episodes?
- Line 285: IPTsc has been already defined above line 45.
- The definition of asymptomatic parasitaemia should be given in the methods.
- Lines 279-281 and lines 282-285 are providing similar information and can be merged.
- It will be easy to understand and follow if authors could specify, IPT with ???? (where ever possible).
- Overall, the discussion needs some improvement, and it is also not clear (lines 330-334) why the control group was treated with SP and AL, when authors say in the method section (lines 87-88) that vitamin C and water were given to the control group.
- What are the good alternatives of Sulphadoxine-Pyrimethamine and Amodiaquine as the presence of molecular markers of resistance to these drugs have been found in Mali (PMID: 27662368)?
